# The potential of the transformer-based survival analysis model, SurvTrace, for predicting recurrent cardiovascular events and stratifying high-risk patients with ischemic heart disease

Hiroki Shinohara[1], Satoshi Kodera[1]*, Yugo Nagae[2], Takashi Hiruma[1], Atsushi Kobayashi[1], Masataka Sato[1], Shinnosuke Sawano[1], Tatsuya Kamon[1], Koichi Narita[1], Kazutoshi Hirose[1], Hiroyuki Kiriyama[1], Akihito Saito[1], Mizuki Miura[1], Shun Minatsuki[1], Hironobu Kikuchi[1], Norifumi Takeda[1], Hiroshi Akazawa[1], Hiroyuki Morita[1], Issei Komuro[1,3]

1 Department of Cardiovascular Medicine, University of Tokyo, Tokyo, Japan, 2 Department of Planning, Information and Management, University of Tokyo, Tokyo, Japan, 3 International University of Health and Welfare, Tokyo, Japan

* koderasatoshi@gmail.com

**Data Availability Statement:** The data contains personal information of each patient and therefore

## Abstract

### Introduction

Ischemic heart disease is a leading cause of death worldwide, and its importance is increasing with the aging population. The aim of this study was to evaluate the accuracy of SurvTrace, a survival analysis model using the Transformer—a state-of-the-art deep learning method—for predicting recurrent cardiovascular events and stratifying high-risk patients. The model's performance was compared to that of a conventional scoring system utilizing real-world data from cardiovascular patients.

### Methods

This study consecutively enrolled patients who underwent percutaneous coronary intervention (PCI) at the Department of Cardiovascular Medicine, University of Tokyo Hospital, between 2005 and 2019. Each patient's initial PCI at our hospital was designated as the index procedure, and a composite of major adverse cardiovascular events (MACE) was monitored for up to two years post-index event. Data regarding patient background, clinical presentation, medical history, medications, and perioperative complications were collected to predict MACE. The performance of two models—a conventional scoring system proposed by Wilson et al. and the Transformer-based model SurvTrace—was evaluated using Harrell's c-index, Kaplan–Meier curves, and log-rank tests.

### Results

A total of 3938 cases were included in the study, with 394 used as the test dataset and the remaining 3544 used for model training. SurvTrace exhibited a mean c-index of 0.72 (95%

cannot be made public. Please contact the Department of Cardiology at the University of Tokyo Hospital (kayo.cho@gmail.com) regarding data sharing. Data sharing is possible after obtaining individual permission from the Ethics Committee of the University of Tokyo Hospital.

**Funding:** Initials of the authors who received each award:H.S. Grant numbers awarded to each author:23K15152 The full name of each funder: Japan Society for the Promotion of Science URL of each funder website:https://kaken.nii.ac.jp/grant/KAKENHI-PROJECT-23K15152/ Did the sponsors or funders play any role in the study design, data collection and analysis, decision to publish, or preparation of the manuscript?: No.

**Competing interests:** The authors have declared that no competing interests exist.

confidence intervals (CI): 0.69–0.76), which indicated higher prognostic accuracy compared with the conventional scoring system's 0.64 (95% CI: 0.64–0.64). Moreover, SurvTrace demonstrated superior risk stratification ability, effectively distinguishing between the high-risk group and other risk categories in terms of event occurrence. In contrast, the conventional system only showed a significant difference between the low-risk and high-risk groups.

## Conclusion

This study based on real-world cardiovascular patient data underscores the potential of the Transformer-based survival analysis model, SurvTrace, for predicting recurrent cardiovascular events and stratifying high-risk patients.

## Introduction

Ischemic heart disease remains the leading cause of death worldwide, despite advancements in treatment modalities and therapeutic technologies [1, 2]. As the population continues to age, improving the prognosis and treatment of ischemic heart disease has become increasingly important. Accurate patient risk stratification is crucial for optimizing treatment, and the effectiveness of scoring systems, such as the Suita score, has been well-documented [3]. Wilson et al. have also reported that scoring models incorporating age and history of catheterization are effective in predicting post-catheterization events [4].

In recent years, rapid advancements in machine learning have shown promise in surpassing conventional methods in patient risk assessment [5, 6]. Beyond standard machine learning survival analysis, new deep learning survival models have been proposed [7]. Specifically, Wang et al. found that a deep learning model known as the "Transformer", which employs an attention mechanism rather than recurrent neural networks or convolutional neural networks, is effective for survival time analysis [8, 9]. The Transformer model has become pivotal in contemporary deep learning, serving as the foundation for systems like ChatGPT [10, 11]. However, no studies have yet assessed the effectiveness of using the Transformer for survival analysis in the cardiovascular field. Therefore, the aim of this study was to compare and validate the accuracy of the novel Transformer-based model against conventional risk scoring model using real-world data from cardiovascular patients.

## Methods

### Study design and participants

This study involved consecutive enrollment of patients who underwent percutaneous coronary intervention (PCI) at the Department of Cardiovascular Medicine, University of Tokyo Hospital, between 2005 and 2019. Within this timeframe, the initial PCI performed at our hospital was designated as the index procedure for each individual patient and used for analysis. Data were accessed and collected for research purposes from October 20, 2022 to December 28, 2022. Information that could identify individual participants was anonymized. A correspondence table was created to ensure that patient information could be accessed after collection, if necessary, while maintaining anonymity. The outcomes of these procedures were evaluated retrospectively. Data on patient background, clinical presentation, medical history, admission medications, perioperative complications, and discharge medications were extracted from the

electronic health records (EHRs) of those who underwent the index PCI. Hypertension was defined as a systolic blood pressure of 140 mmHg or higher upon admission, a diastolic blood pressure of 90 mmHg or higher upon admission, or ongoing treatment with antihypertensive medications. Diabetes mellitus was defined by a hemoglobin A1c level $\geq$6.5% upon admission or ongoing treatment with either insulin or oral hypoglycemic agents. Dyslipidemia was defined as a low-density lipoprotein cholesterol level $\geq$140 mg/dL upon admission, a high-density lipoprotein cholesterol < 40 mg/dL upon admission, triglycerides $\geq$150 mg/dL upon admission, or ongoing use of dyslipidemia medications. Chronic kidney disease was defined as patients with an eGFR <60 mL/minute/1.73 m$^2$, calculated using the Modification of Diet in Renal Disease (MDRD) equation [12] and serum creatinine levels upon admission modified by Japanese coefficients.

Missing data constituted 1.0% of all variables in the total dataset. These missing values were addressed using the multiple imputation method [13]. This technique substituted missing data points with a set of plausible alternatives, thereby generating multiple complete datasets for analysis. Each dataset was individually analyzed, and the results were then aggregated to produce a single, comprehensive result. In this study, we used Python to generate five pseudo-complete datasets, applying multiple imputations using the Bayesian Ridge method (S1 File).

To improve model interpretability and minimize multicollinearity, Pearson's correlation coefficient was used to assess the correlation among explanatory variables. Any variable exhibiting a Pearson's correlation coefficient exceeding 0.90 was omitted from the set of explanatory variables used for model training [14]. In cases where two features were highly correlated, the one with the greater overall correlation to all features was eliminated [14]. During the preprocessing phase, all continuous variables were standardized to have a mean value of 0 and a standard deviation of 1.

The endpoint consisted of a composite of major adverse cardiovascular events (MACE), including cardiac death, acute coronary syndrome, cerebrovascular event, and hospitalization for heart failure [4]. EHRs were used to collect data on these outcomes, as well as the period until their occurrence, for up to two years following the index procedure. Cardiac death was defined as death from acute myocardial infarction, ventricular arrhythmia, or heart failure [15]. Acute coronary syndrome was defined as nonfatal myocardial infarction or unstable angina [15]. Nonfatal myocardial infarction was defined as persistent angina accompanied by new ECG abnormalities and elevated cardiac biomarkers [15]. Unstable angina pectoris was defined as an extended episode of resting ischemic symptoms (typically exceeding 10 minutes) or a lowering of the activity threshold that induced accelerated chest pain, necessitating an unscheduled medical visit and an overnight stay—usually within 24 hours of the most recent symptoms—while not fulfilling myocardial infarction cardiac biomarker criteria [16]. Cerebrovascular events were defined as either cerebral hemorrhage or cerebral infarction. Survival time analyses were conducted on these outcomes until the respective dates of event onset. To compare the prognostic accuracy of the novel Transformer-based model with that of the conventional risk scoring model, the c-index was employed [17]. Subsequently, the risk stratification capabilities of each model were assessed by computing risk scores for every patient using the trained models. Patients in the test set were classified into high-, intermediate-, and low-risk score groups [18] and evaluated through Kaplan–Meier survival curves [19] and log-rank tests [20].

The impact of explanatory variables on outcomes was assessed using Shapley additive explanations (SHAP) [21]. An algorithmic evaluation method rooted in game theory, SHAP uses Shapley scores to estimate the contribution of each explanatory variable to the model's prediction.

To assess the robustness of our findings, we performed three distinct sensitivity analyses: first, by omitting missing values; second, by adjusting the percentage of test sets; and third, by excluding patients with a history of PCI. This study was conducted in accordance with the revised Declaration of Helsinki and received approval from the institutional review board of the University of Tokyo Hospital (2021238NI-(2)). Informed consent was obtained in the form of an opt-out on a website.

## Modeling

To evaluate the predictive accuracy of MACE, we utilized the scoring system proposed by Wilson et al. [4] and SurvTrace, which is based on a model that uses a Transformer architecture [8].

The scoring system proposed by Wilson et al. serves as a predictive model for recurrent cardiovascular disease and incorporates variables such as age, smoking history, history of diabetes or heart failure, body mass index, number of diseased vessels, and history of statin or aspirin therapy. For the purposes of this study, it was defined as a conventional scoring model. SurvTrace is an alternative survival time analysis model that employs a Transformer, a specific deep learning technique. Using an attention mechanism, this model enables efficient calculation of the effect of each variable on survival time. All computational models were implemented using Python and executed on an Nvidia Tesla A-100 80GB graphics processing unit.

For data partitioning, 90% of the total dataset was randomly selected to constitute the training set. Subsequently, 25% of this training set was randomly allocated for validation during the model training process. The remaining 10% of the data, which was not included in the training set, served as a test set for assessing the accuracy of the trained models. Throughout the training process, Optuna, an advanced framework for hyperparameter optimization tailored for machine learning, was employed to fine-tune the model's hyperparameters [22]. S2 File shows the SurvTrace execution code used.

## Statistical analysis

Five pseudo-complete datasets were generated through the application of multiple imputation techniques to address missing values. The model's accuracy was then calculated based on these five datasets. To synthesize the findings, the five accuracy estimates derived from each model were integrated using Rubin's rules, facilitating a comparison of model performance [23].

For continuous variables, measurements were expressed as either mean (± standard deviation) or median (first and third quartiles), while categorical variables were reported as counts and frequencies (%).

The models' prognostic accuracy was assessed using Harrell's c-index [18]. Additionally, the risk stratification capabilities of each model were assessed through Kaplan–Meier curves [20] and log-rank tests [21]. The $p$ value threshold for significance was set at $<0.05$. All statistical analyses were performed using Python 3.7.

## Results

Between January 1, 2005, and December 31, 2019, a total of 3938 first-time PCIs were performed in our hospital. Of these, 394 were designated as the test dataset, while the remaining 3544 cases were used for model training (Fig 1). Among the patient information data collected from the EHRs at the University of Tokyo Hospital, 171 explanatory variables were used. Table 1 outlines the baseline characteristics of the key explanatory variables. The training dataset contained a significantly higher number of patients with a history of previous PCI

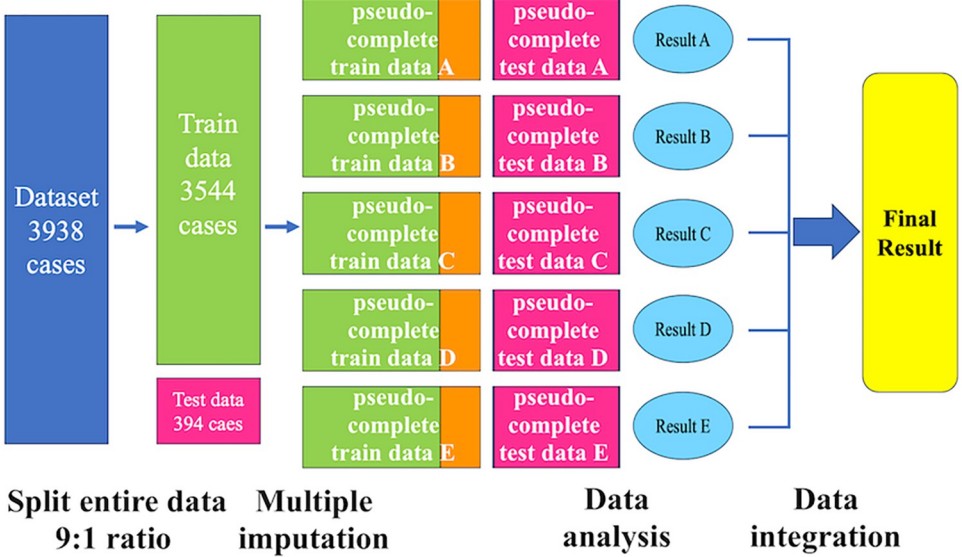

**Fig 1. Study flowchart.**

compared with the test dataset. During the observation period, 683 subjects (17.3%) were lost to follow-up, including 610 cases in the training dataset and 73 cases in the test dataset.

The c-index of SurvTrace outperformed that of the conventional scoring system, registering a mean c-index of 0.72 (95% confidence interval: 0.69–0.76), as opposed to a mean c-index of 0.64 (95% confidence interval: 0.64–0.64) for the conventional scoring system (Table 2, Fig 2). Fig 3 illustrates the learning curve of SurvTrace during its training process. The most accurate training model from among all trained risk prediction models, along with its dataset, was used to evaluate risk stratification capabilities. While the conventional scoring system showed that the low-risk group experienced significantly fewer events compared with the high-risk group, it did not show a significant difference between the intermediate-risk group and the other patient groups (Fig 4). In contrast, SurvTrace revealed that the high-risk group had a significantly higher number of events than the other groups (Fig 4).

Fig 5 presents the SHAP result, indicating that SurvTrace highlighted the influence of pre-existing conditions, such as a history of chronic heart failure.

In the first sensitivity analysis, cases with missing values were excluded from both training and test datasets. Post-exclusion, the training dataset comprised 2137 cases, and the test dataset contained 254 cases. The c-index for SurvTrace was 0.71, compared with 0.66 for the conventional scoring system. The second sensitivity analysis involved adjusting the proportion of the test dataset to 20%. Following this modification, the analysis was performed using one of the five pseudo-complete datasets generated by the multiple imputation method, including both training and test datasets. This adjustment yielded a c-index of 0.68 for SurvTrace and 0.66 for the conventional scoring system. In the final sensitivity analysis, after excluding patients with a history of PCI from one of the five pseudo-complete training and test datasets, the c-index for SurvTrace was 0.69, compared with 0.63 for the conventional scoring system.

This figure illustrates the flowchart of the study. Initially, all data were split into training and test datasets at a 9:1 ratio. To address missing values, multiple imputation was applied to both datasets, generating five pseudo-complete datasets for each. A separate 25% segment of the training dataset was reserved for validation. Subsequently, survival analysis was performed on each pseudo-complete dataset, and the c-index was calculated. Finally, Rubin's rules were

**Table 1. Baseline characteristics of key explanatory variables.**

| Baseline characteristics | Overall N = 3938 | Training data, N = 3,544 | Test data, N = 394 | p-value |
|---|---|---|---|---|
| **Patient characteristics** | | | | |
| Age | 68.4 ± 10.4 | 68.5 ± 10.3 | 67.5 ± 11.0 | 0.10 |
| Male, n (%) | 3,062 / 3,938 (77.8) | 2,748 / 3,544 (77.5) | 314 / 394 (79.7) | 0.36 |
| BMI, kg/m2 | 24.3 ± 3.9 | 24.3 ± 3.9 | 24.2 ± 3.9 | 0.83 |
| Smoking history, n (%) | 2,546 / 3,920 (64.9) | 2,287 / 3,528 (64.8) | 259 / 392 (66.1) | 0.66 |
| **History of cardiac surgery** | | | | |
| PCI, n (%) | 653 / 3,932 (16.6) | 605 / 3,538 (17.1) | 48 / 394 (12.2) | **0.016** |
| CABG, n (%) | 297 / 3,935 (7.5) | 266 / 3,541 (7.5) | 31 / 394 (7.9) | 0.88 |
| **Medical history** | | | | |
| MI, n (%) | 443 / 3,934 (11.3) | 405 / 3,540 (11.4) | 38 / 394 (9.6) | 0.32 |
| CHF, n (%) | 279 / 3,933 (7.1) | 254 / 3,539 (7.2) | 25 / 394 (6.3) | 0.61 |
| Af, n (%) | 350 / 3,938 (8.9) | 318 / 3,544 (9.0) | 32 / 394 (8.1) | 0.64 |
| HTN, n (%) | 3,358 / 3,930 (85.4) | 3,024 / 3,537 (85.5) | 334 / 393 (85.0) | 0.84 |
| DM, n (%) | 2,027 / 3,932 (51.6) | 1,811 / 3,539 (51.2) | 216 / 393 (55.0) | 0.17 |
| DL, n (%) | 3,125 / 3,931 (79.5) | 2,803 / 3,538 (79.2) | 322 / 393 (81.9) | 0.23 |
| CKD, n (%) | 1,642 / 3,932 (41.8) | 1,476 / 3,539 (41.7) | 166 / 393 (42.2) | 0.88 |
| HD, n (%) | 253 / 3,932 (6.4) | 227 / 3,539 (6.4) | 26 / 393 (6.6) | 0.96 |
| Peripheral arterial disease, n (%) | 482 / 3,929 (12.3) | 445 / 3,536 (12.6) | 37 / 393 (9.4) | 0.083 |
| **Drugs at admission** | | | | |
| Antiplatelet drugs, n (%) | 2,666 / 3,911 (68.2) | 2,408 / 3,522 (68.4) | 258 / 389 (66.3) | 0.44 |
| Statins, n (%) | 1,968 / 3,912 (50.3) | 1,763 / 3,523 (50.0) | 205 / 389 (52.7) | 0.35 |
| Beta blockers, n (%) | 1,083 / 3,911 (27.7) | 973 / 3,522 (27.6) | 110 / 389 (28.3) | 0.83 |
| ACEIs, n (%) | 435 / 3,911 (11.1) | 399 / 3,522 (11.3) | 36 / 389 (9.3) | 0.25 |
| Diuretics, n (%) | 608 / 3,911 (15.5) | 550 / 3,522 (15.6) | 58 / 389 (14.9) | 0.77 |
| **Cardiac disease at admission** | | | | |
| Stable Angina, n (%) | 2,785 / 3,938 (70.7) | 2,507 / 3,544 (70.7) | 278 / 394 (70.6) | 0.99 |
| ACS, n (%) | 1,153 / 3,938 (29.3) | 1,037 / 3,544 (29.3) | 116 / 394 (29.4) | 0.99 |
| Killip's classification ≧III, n (%) | 189 / 3,938 (4.8) | 169 / 3,544 (4.8) | 20 / 394 (5.1) | 0.88 |
| Peak CK, U/L | 112 (77–173) | 112 (77–173) | 114 (79–181) | 0.77 |
| **Outcome in this study** | | | | |
| MACE, n (%) | 315 / 3,938 (8.0) | 280 / 3,544 (7.9) | 35 / 394 (8.9) | 0.56 |

Values are shown as n (%), mean ± standard deviation, or median (first and third quartiles).

BMI = body mass index, PCI = percutaneous coronary intervention, CABG = coronary artery bypass graft, MI = myocardial infarction, CHF = chronic heart failure, Af = atrial fibrillation, HTN = hypertension, DM = diabetes mellitus, DL = dyslipidemia, CKD = chronic kidney disease, HD = hemodialysis, ACEI = angiotensin-converting enzyme inhibitor, ACS = acute coronary syndrome, CK = creatine kinase, MACE = major adverse cardiovascular events

used to integrate the c-index values from each dataset to compute the final result. In the figure, yellow-green represents the data used for training the model, orange represents the validation data, and pink represents the data used for testing post-training.

**Table 2. Results for both models.**

| | Pseudo-complete DatasetA Cindex | Pseudo-complete DatasetB Cindex | Pseudo-complete DatasetC Cindex | Pseudo-complete DatasetD Cindex | Pseudo-complete DatasetE Cindex | Mean Cindex | Standard Error | 95%CI Low | 95%CI High |
|---|---|---|---|---|---|---|---|---|---|
| Conventional scoring system | 0.64 | 0.64 | 0.64 | 0.64 | 0.64 | 0.64 | 0.00072 | 0.64 | 0.64 |
| SurvTrace | 0.74 | 0.71 | 0.71 | 0.73 | 0.72 | 0.72 | 0.015 | 0.69 | 0.76 |

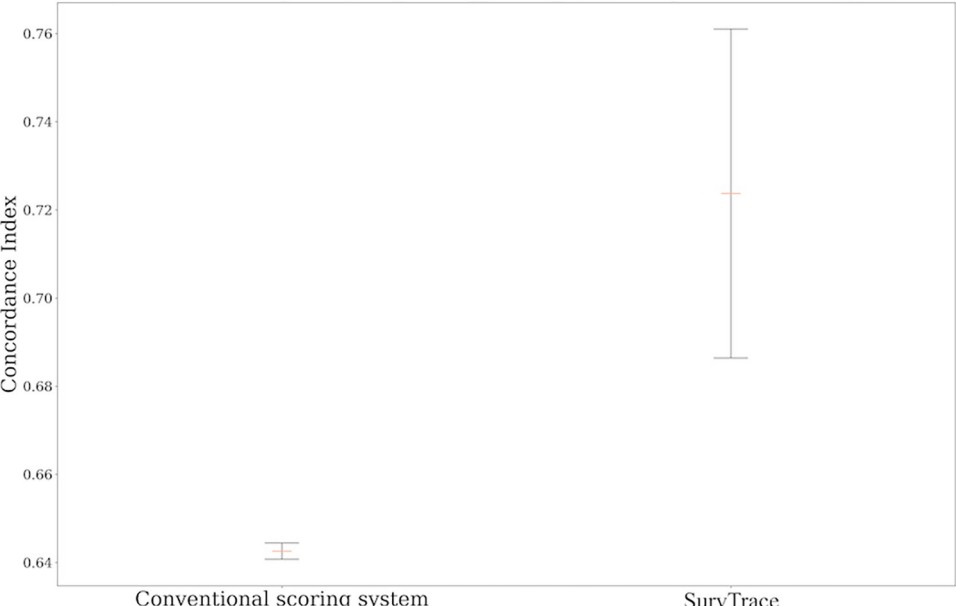

**Fig 2. C-indices of the models.** This figure shows the c-index for both the conventional scoring system and SurvTrace. The upper and lower black lines represent the upper and lower limits of the 95% confidence intervals, respectively. The orange line shows the mean c-index value calculated from five pseudo-complete datasets.

## Discussion

This study demonstrated that SurvTrace, a predictive model using the Transformer deep learning algorithm, was effective in predicting recurrent cardiovascular events in patients with ischemic heart disease based on real-world clinical data. Compared with conventional scoring system, SurvTrace not only demonstrated superior accuracy in event prediction but also showed an improved ability to stratify high-risk patients.

The Transformer-based SurvTrace model demonstrated significantly higher prediction accuracy for recurrent cardiovascular events in patients with ischemic heart disease, using real-world clinical data, than did conventional scoring system. SurvTrace also demonstrated a

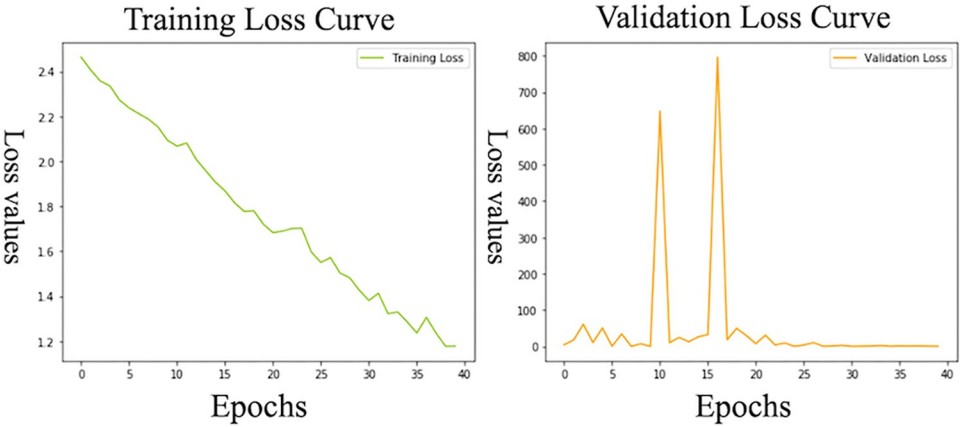

**Fig 3. Learning curve of SurvTrace during the training process.** This figure illustrates the variation in the loss function over the course of the training process. The left panel shows the fluctuations in loss values for the training dataset, while the right panel shows these changes for the validation dataset.

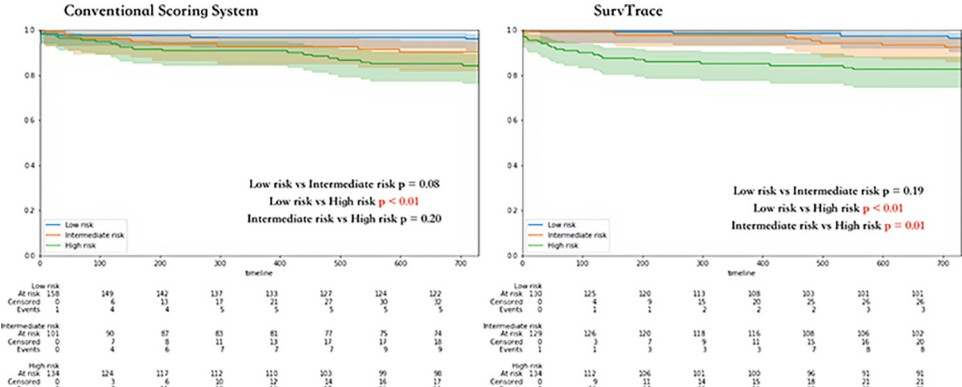

**Fig 4. Kaplan–Meier curves of the models.** This figure shows the Kaplan–Meier curves generated by both the conventional scoring model and SurvTrace. The blue lines represent the Kaplan–Meier curve for the low-risk group as stratified by risk scores from both models. Similarly, the orange and green lines represent the curves for the intermediate- and high-risk groups, respectively. The translucent segments of each line indicate the 95% confidence interval.

significantly greater capacity for high-risk patient stratification relative to conventional scoring system. The model maintained its superior performance across a range of sensitivity analyses, which included the exclusion of missing values from the training and test datasets, modification of the test set percentages, and the exclusion of patients with a history of PCI. These results are consistent with previous studies that have underscored the superiority of machine learning and deep learning algorithms over conventional scoring systems [6, 18]. The high accuracy of these advanced models is likely attributed to their ability to identify complex patterns among explanatory variables, a feature not present in conventional methods. Typically, conventional scoring systems rely on linear models, selecting only statistically significant explanatory variables. Such models necessitate explicit definitions of relationships between explanatory variables to account for any interactions, thereby increasing model complexity and raising concerns about multicollinearity and overfitting as the number of variables grows.

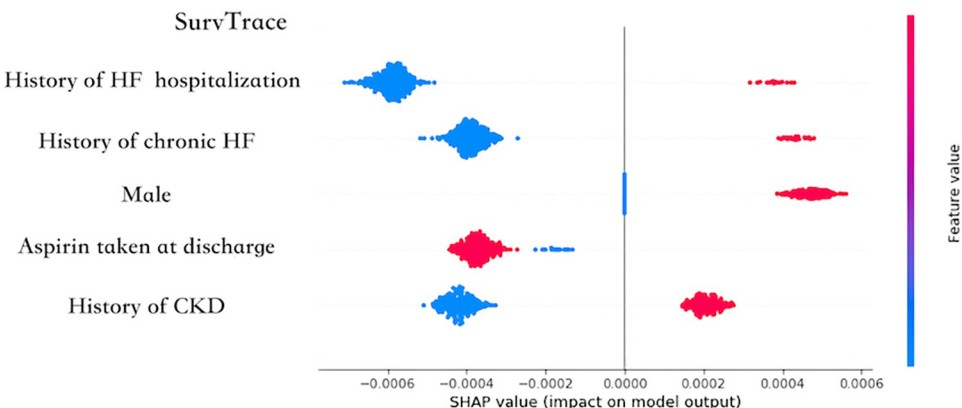

**Fig 5. Summary plot of SurvTrace.** This figure illustrates the Shapley additive explanations (SHAP) of SurvTrace. The horizontal axis indicates the impact on the model's prediction, with points situated to the right representing a higher risk of future major adverse cardiovascular events (MACE) compared with points on the left. The vertical axis indicates the importance of the explanatory variables. In this model, a history of hospitalization for heart failure (HF) exerts the greatest impact on predicting the risk of future MACE events. The color of each dot indicates the high or low status within each variable; for example, in the "History of HF Hospitalization" column, red indicates that the patient has a history of HF hospitalization, while blue indicates no such history.

In contrast, the Transformer algorithm can directly incorporate multiple explanatory variables into its models, capturing nonlinear relationships and complex interactions among them without the need for explicit definitions. In this study, while the conventional scoring system incorporated only important variables such as age, gender, and medical history, SurvTrace used all 171 explanatory variables. This comprehensive approach to feature inclusion may contribute to its higher predictive accuracy.

The Transformer model's ability to stratify high-risk group more accurately than conventional scoring system has important implications for managing patients with ischemic heart disease in real-world clinical practice. Moreover, the alignment of our SHAP results with prior findings further underscores the robustness and validity of our study's outcomes. The enhanced risk stratification capabilities of the Transformer model could potentially improve clinical decision making and assist physicians in tailoring treatment plans for individual patients [24]. Recent advancements have introduced large language models capable of automatically extracting structured data from electronic medical records [25, 26]. Using these language models enables automated survival time analysis and future risk stratification based on individual patient records, offering a more personalized treatment approach that may potentially enhance intervention effectiveness and improve patient outcomes.

This study has several limitations that warrant consideration. First, our research relied on a dataset from a single institution, making it susceptible to potential selection bias. Future studies should address institution-specific biases by expanding and validating the diversity of the patient population through multicenter studies. Second, the sample size was relatively modest, comprising 3938 patients. In general, deep learning models require larger datasets to achieve high levels of accuracy; therefore, our sample size may have been insufficient. Third, although this study demonstrated the superiority of the Transformer model over conventional scoring system, it should be noted that the model used was specific to this study. Other Transformer models not evaluated in this study may yield different results. Fourth, this study was retrospective in nature, with events meticulously tracked in the EHRs. Despite this thorough tracking, some events might have been overlooked as a result of patients relocating or transferring to other hospitals, potentially leading to selection bias. To mitigate this issue, future prospective studies employing survival analysis with the Transformer model are necessary. Lastly, missing values in the dataset were handled using multiple imputation methods to facilitate the Transformer model's application. These imputed values could introduce bias, especially for the Transformer model, as deep learning models are known to be sensitive to data noise.

## Conclusion

This study demonstrated that a survival analysis model using Transformer, a state-of-the-art deep learning method, was significantly more accurate than the conventional scoring system in predicting recurrent cardiovascular events and stratifying high-risk patients using real-world clinical data. Additional research is warranted to further optimize the performance of deep learning models for more effective risk stratification and management of patients with ischemic heart disease.

## Supporting information

**S1 File. Multiple imputation method execution file.** This file contains the code to execute the multiple imputation method in Python.
(DOCX)

**S2 File. SurvTrace execution file.** This file contains the code to execute SurvTrace in Python. (DOCX)

## Acknowledgments

We thank Phoebe Chi, MD, from Edanz (https://jp.edanz.com/ac) for editing a draft of this manuscript.

## Author Contributions

**Conceptualization:** Hiroki Shinohara, Satoshi Kodera.

**Data curation:** Hiroki Shinohara, Yugo Nagae, Takashi Hiruma, Atsushi Kobayashi, Masataka Sato, Shinnosuke Sawano, Tatsuya Kamon, Koichi Narita, Kazutoshi Hirose, Hiroyuki Kiriyama, Akihito Saito, Mizuki Miura, Shun Minatsuki, Hironobu Kikuchi.

**Formal analysis:** Hiroki Shinohara.

**Funding acquisition:** Hiroki Shinohara.

**Project administration:** Satoshi Kodera.

**Supervision:** Satoshi Kodera, Norifumi Takeda, Hiroshi Akazawa, Hiroyuki Morita, Issei Komuro.

**Writing – original draft:** Hiroki Shinohara.

**Writing – review & editing:** Satoshi Kodera.

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
