## [Decision Letter · Decision Letter 0]

14 Dec 2023

PONE-D-23-35647The Potential of the Transformer-based Survival Analysis Model, SurvTrace, for Predicting Recurrent Cardiovascular Events and Stratifying High-risk Patients with Ischemic Heart DiseasePLOS ONE

Dear Dr. Kodera,

Thank you for submitting your manuscript to PLOS ONE. After careful consideration, we feel that it has merit but does not fully meet PLOS ONE’s publication criteria as it currently stands. Therefore, we invite you to submit a revised version of the manuscript that addresses the points raised during the review process.

**ACADEMIC EDITOR: **

The manuscript is interesting but will require further reworking and a major revision.<o:p></o:p>

While they recognize the potential interest of the subject studied, the reviewers raised a number of important issues that need to be properly addressed.

We look forward to receiving your revised manuscript.

Kind regards,

Marcelo Arruda Nakazone, M.D., Ph.D.

Academic Editor

PLOS ONE

Reviewers' comments:

Reviewer's Responses to Questions

**Comments to the Author**

1. Is the manuscript technically sound, and do the data support the conclusions?

Reviewer #1: Yes

Reviewer #2: Partly

2. Has the statistical analysis been performed appropriately and rigorously? 

Reviewer #1: I Don't Know

Reviewer #2: Yes

3. Have the authors made all data underlying the findings in their manuscript fully available?

Reviewer #1: Yes

Reviewer #2: No

4. Is the manuscript presented in an intelligible fashion and written in standard English?

Reviewer #1: Yes

Reviewer #2: Yes

5. Review Comments to the Author

Reviewer #1: The manuscript presents a retrospective cohort study aimed at assessing the precision of a survival analysis model for predicting cardiovascular events. It would be beneficial for the authors to align their reporting with the Tripod guidelines to enhance the clarity and reproducibility of their research.

In terms of methodology, the manuscript would benefit from explicitly stating any exclusion criteria that were applied to the study population. Furthermore, a rationale should be provided for the allocation of 10% of the dataset for validation purposes, with the remaining 90% utilized for the development and testing of the model.

Regarding the statistical analysis, it is crucial for the authors to specify the exact multiple imputation method employed. It is also advisable to make the Python code used for the analysis available, as this would greatly aid in the transparency and replicability of the study. Lastly, performing a sensitivity analysis to gauge the influence of missing data on the study's outcomes—comparing results from complete case analysis with those from multiple imputation—would significantly strengthen the findings.

Reviewer #2: I am grateful for the opportunity to review this manuscript evaluating the accuracy of a survival analysis algorithm that's based on the transformer deep learning model in predicting the development of major adverse cardiovascular events among patients who underwent percutaneous coronary intervention. The manuscript is well-written and easy to digest. However, I note some important comments below:

1. This analysis is based on data from a single healthcare system. Patients were not actively followed prospectively to evaluate study outcomes. Therefore, if a patient develops the outcome but they were not admitted at the hospital, they will likely be adjudged as not having the outcome because it was not observed in the hospital. This may introduce a huge amount of bias in outcome ascertainment. The authors need to develop some measures that will reduce this bias. For example, the authors can limit their analysis to established patients of the hospital (i.e. patients who are in a way loyal to the hospital). Established patients can be defined based on certain plausible criteria or based on previous studies.

2. Furthermore, the authors need to address or discuss how demographic changes such as emigration or even a patient moving away from the hospital service area can affect their ability to track outcome development among the patients.

3. Was there any attempt to ensure that the analysis cohort is made up of patients who are getting PCI for the first time? Otherwise, can the authors provide data on the proportion of included patients who have already received a previous PCI before the index event (per this study's criteria)?

4. In line 87, the authors mentioned that Any variable exhibiting a Pearson’s correlation coefficient exceeding 0.90 was omitted from set of explanatory variables used for model training. How did the authors decide which of the two variables with correlation >0.9 between them is dropped from the analysis?

5. How did the authors arrive at the decision to use 90% of the dataset for training? Was there any form of sensitivity analysis done to arrive at the optimal data splitting ratio?

6. The authors have not presented the measures of the fitness of the deep learning models (For example, the test and training accuracy by epoch)

7. Model hyperparameter tuning were not discussed. How does a reader know that the final model is the best model?

8. The authors should ideally provide the codes used for their analysis.

6. PLOS authors have the option to publish the peer review history of their article (what does this mean?). If published, this will include your full peer review and any attached files.

Reviewer #1: **Yes: **Ang Yee Gary

Reviewer #2: No

---

## [Author Response · Author response to Decision Letter 0]

2 Feb 2024

Reviewer #1: The manuscript presents a retrospective cohort study aimed at assessing the precision of a survival analysis model for predicting cardiovascular events. It would be beneficial for the authors to align their reporting with the Tripod guidelines to enhance the clarity and reproducibility of their research.

We would like to express our sincere gratitude for your thorough and insightful review of our manuscript. Your constructive comments and suggestions are greatly appreciated and have provided valuable guidance in improving the quality and clarity of our research. We recognize the importance of adhering to established guidelines and methodological rigor, and we are grateful for the opportunity to improve our work based on your expert feedback.

After reviewing the Tripod guidelines and considering your comments, we have revised our manuscript to better meet these standards. This revision focused on providing a more detailed and transparent presentation of our methodology, data analysis, and results. By aligning our report with the Tripod guidelines, we have improved both the clarity and replicability of our research.

・In terms of methodology, the manuscript would benefit from explicitly stating any exclusion criteria that were applied to the study population. 

Thank you for your valuable comments regarding clarification of the exclusion criteria applied to our study population. As this study enrolled all consecutive patients who underwent percutaneous coronary intervention (PCI) at our institution, we did not apply any specific exclusion criteria. To clarify this point, we have revised the manuscript on page 4, line 58 as follows: " This study involved consecutive enrollment of patients who underwent percutaneous coronary intervention (PCI) at the Department of Cardiovascular Medicine, University of Tokyo Hospital, between 2005 and 2019. Within this timeframe, the initial PCI performed at our hospital was designated as the index procedure for each individual patient and used for analysis." We believe this revision provides a clearer understanding of our study design and patient selection process.

・Furthermore, a rationale should be provided for the allocation of 10% of the dataset for validation purposes, with the remaining 90% utilized for the development and testing of the model.

Thank you for highlighting the need to provide a detailed rationale for the allocation of our dataset into training, validation, and test sets. In our study, we allocated 10% of the dataset as a test set and divided the remaining 90% into training and validation sets at a 3:1 ratio. This decision was guided by the requirements of the Transformer model used in our analysis, which requires a substantial amount of data for effective training while ensuring an adequate amount for unbiased testing.

To improve the clarity of our methodology, we have updated Figure 1. This revised figure now includes a note on the validation data set and illustrates the data partitioning process. The legend for Figure 1, found on page 11, line 198, has been revised to read: “This figure illustrates the flowchart of the study. Initially, all data were split into training and test datasets at a 9:1 ratio. To address missing values, multiple imputation was applied to both datasets, generating five pseudo-complete datasets for each. A separate 25% segment of the training dataset was reserved for validation. Subsequently, survival analysis was performed on each pseudo-complete dataset, and the c-index was calculated. Finally, Rubin’s rules were used to integrate the c-index values from each dataset to compute the final result. In the figure, yellow-green represents the data used for training the model, orange represents the validation data, and pink represents the data used for testing post-training.”

In addition, to thoroughly address concerns about the impact of the percentage of test data on our results, we performed extensive sensitivity analyses with different allocations of the test set, including a scenario where 20% of the dataset was used for testing. The methodology and results of these analyses are carefully detailed in our new manuscript. Specifically, on page 7, line 120, the text reads: "To assess the robustness of our findings, we performed three distinct sensitivity analyses: first, by omitting missing values; second, by adjusting the percentage of test sets; and third, by excluding patients with a history of PCI. " This section outlines the steps taken in our sensitivity analyses. Further, the results of these analyses are elaborated on page 11, line 188, with the statement: “The second sensitivity analysis involved adjusting the proportion of the test dataset to 20%. Following this modification, the analysis was performed using one of the five pseudo-complete datasets generated by the multiple imputation method, including both training and test datasets. This adjustment yielded a c-index of 0.68 for SurvTrace and 0.66 for the conventional scoring system.”

・Regarding the statistical analysis, it is crucial for the authors to specify the exact multiple imputation method employed. It is also advisable to make the Python code used for the analysis available, as this would greatly aid in the transparency and replicability of the study. Lastly, performing a sensitivity analysis to gauge the influence of missing data on the study's outcomes—comparing results from complete case analysis with those from multiple imputation—would significantly strengthen the findings.

We very much appreciate your comments on our statistical analysis, in particular the need to specify the multiple imputation method used and the importance of transparency and replicability in our study.

First, we have addressed your comment about specifying the multiple imputation method. We have updated our manuscript on page 5, line 85, to clearly state: “In this study, we used Python to generate five pseudo-complete datasets, applying multiple imputations using the Bayesian Ridge method (S1 File).” In addition, to increase transparency and facilitate replication of our study, we have included the Python script used for the analysis as S1 file.

Furthermore, per your suggestion for a sensitivity analysis to assess the influence of missing data on the results of our study, we have performed such analyses. On page 7, line 120, the following has been added: “To assess the robustness of our findings, we performed three distinct sensitivity analyses: first, by omitting missing values; second, by adjusting the percentage of test sets; and third, by excluding patients with a history of PCI.” In addition, the results of the first sensitivity analysis are elaborated on page 11, line 185: “In the first sensitivity analysis, cases with missing values were excluded from both training and test datasets. Post-exclusion, the training dataset comprised 2137 cases, and the test dataset contained 254 cases. The c-index for SurvTrace was 0.71, compared with 0.66 for the conventional scoring system.” This analysis confirms that the overall trends in our results remain consistent even when accounting for missing data.

 

Reviewer #2: I am grateful for the opportunity to review this manuscript evaluating the accuracy of a survival analysis algorithm that's based on the transformer deep learning model in predicting the development of major adverse cardiovascular events among patients who underwent percutaneous coronary intervention. The manuscript is well-written and easy to digest. However, I note some important comments below:

We would like to express our deepest gratitude for your time and effort in reviewing our manuscript. We are particularly grateful for your positive comments on the clarity and readability of our work. It is encouraging to hear that our manuscript, which evaluates the accuracy of a survival analysis algorithm based on the Transformer deep learning model in predicting major adverse cardiovascular events in patients undergoing percutaneous coronary intervention, has been well received. We appreciate your constructive feedback and are committed to addressing your concerns and suggestions.

1. This analysis is based on data from a single healthcare system. Patients were not actively followed prospectively to evaluate study outcomes. Therefore, if a patient develops the outcome but they were not admitted at the hospital, they will likely be adjudged as not having the outcome because it was not observed in the hospital. This may introduce a huge amount of bias in outcome ascertainment. The authors need to develop some measures that will reduce this bias. For example, the authors can limit their analysis to established patients of the hospital (i.e. patients who are in a way loyal to the hospital). Established patients can be defined based on certain plausible criteria or based on previous studies.

We sincerely appreciate your critical observation regarding the potential bias in outcome ascertainment due to the retrospective nature of our study and the reliance on data from a single healthcare system. As you rightly point out, there is indeed a possibility that not all events were captured, particularly if they occurred outside the hospital or if patients were not admitted to our hospital for subsequent care.

In response to your insightful suggestion, we have made significant additions to our manuscript to address this issue. Specifically, we have added a statement on page 10, line 169, that reads: “During the observation period, 683 subjects (17.3%) were lost to follow-up, including 610 cases in the training dataset and 73 cases in the test dataset.” This addition is intended to provide transparency regarding the extent of follow-up within our study population.

In addition, we have further acknowledged this limitation and the potential for selective bias in our results on page 19, line 299, in the Limitations section: “Fourth, this study was retrospective in nature, with events meticulously tracked in the EHRs. Despite this thorough tracking, some events might have been overlooked as a result of patients relocating or transferring to other hospitals, potentially leading to selective bias. To mitigate this issue, future prospective studies employing survival analysis with the Transformer model are necessary.”

We believe these changes will help provide a more comprehensive understanding of the scope and limitations of our study.

2. Furthermore, the authors need to address or discuss how demographic changes such as emigration or even a patient moving away from the hospital service area can affect their ability to track outcome development among the patients.

Thank you for your important observation regarding the impact of demographic changes, such as migration or patients moving out of the hospital's service area, on our ability to track patient outcomes. As mentioned in response to point 1, we acknowledge that as a single-center, retrospective study, there is indeed a risk of selective bias.

Given your comment, we extended the limitations section of our manuscript as mentioned above. On page 19, line 299, we added: “Fourth, this study was retrospective in nature, with events meticulously tracked in the EHRs. Despite this thorough tracking, some events might have been overlooked as a result of patients relocating or transferring to other hospitals, potentially leading to selective bias. To mitigate this issue, future prospective studies employing survival analysis with the Transformer model are necessary.”

3. Was there any attempt to ensure that the analysis cohort is made up of patients who are getting PCI for the first time? Otherwise, can the authors provide data on the proportion of included patients who have already received a previous PCI before the index event (per this study's criteria)?

Thank you for raising the important question regarding the inclusion criteria of our patient cohort, especially concerning those who have previously undergone percutaneous coronary intervention (PCI).

In this study, we consecutively enrolled patients who underwent PCI at our hospital. For purposes of analysis, the first PCI performed at our institution was considered the index PCI for each patient. Therefore, our analysis cohort included patients with a history of PCI prior to the index PCI at our institution. Page 4, line 58 reads “This study involved consecutive enrollment of patients who underwent percutaneous coronary intervention (PCI) at the Department of Cardiovascular Medicine, University of Tokyo Hospital, between 2005 and 2019. Within this timeframe, the initial PCI performed at our hospital was designated as the index procedure for each individual patient and used for analysis.”

In response to your insightful suggestion, we have conducted a sensitivity analysis to evaluate the impact of including patients with a prior history of PCI. On page 7, line 120, the following has been added: “To assess the robustness of our findings, we performed three distinct sensitivity analyses: first, by omitting missing values; second, by adjusting the percentage of test sets; and third, by excluding patients with a history of PCI.” In addition, the results of the sensitivity analysis are elaborated on page 11, line 192: “In the final sensitivity analysis, after excluding patients with a history of PCI from one of the five pseudo-complete training and test datasets, the c-index for SurvTrace was 0.69, compared with 0.63 for the conventional scoring system.”

4. In line 87, the authors mentioned that Any variable exhibiting a Pearson’s correlation coefficient exceeding 0.90 was omitted from set of explanatory variables used for model training. How did the authors decide which of the two variables with correlation >0.9 between them is dropped from the analysis?

We apologize for any lack of clarity in our manuscript regarding the methodology for handling highly correlated variables. We appreciate your question, which has highlighted an area in need of further explanation.

To address the issue of multicollinearity in our dataset, we referred to the methodology outlined in reference 14. Based on this, when two features exhibited a Pearson’s correlation coefficient exceeding 0.90, we removed the one with the highest overall correlation to all other features. 

To provide greater clarity on this aspect, we have revised our manuscript accordingly. On page 6, line 91, we have added the following clarification: "In cases where two features were highly correlated, the one with the greater overall correlation to all features was eliminated "

5. How did the authors arrive at the decision to use 90% of the dataset for training? Was there any form of sensitivity analysis done to arrive at the optimal data splitting ratio?

Thank you for highlighting the need to provide a detailed rationale for the allocation of our dataset into training, validation, and test sets. In our study, we allocated 10% of the dataset as a test set and divided the remaining 90% into training and validation sets at a 3:1 ratio. This decision was guided by the requirements of the Transformer model used in our analysis, which requires a substantial amount of data for effective training while ensuring an adequate amount for unbiased testing.

To improve the clarity of our methodology, we have updated Figure 1. This revised figure now includes a note on the validation data set and illustrates the data partitioning process. The legend for Figure 1, found on page 11, line 198, has been revised to read: “This figure illustrates the flowchart of the study. Initially, all data were split into training and test datasets at a 9:1 ratio. To address missing values, multiple imputation was applied to both datasets, generating five pseudo-complete datasets for each. A separate 25% segment of the training dataset was reserved for validation. Subsequently, survival analysis was performed on each pseudo-complete dataset, and the c-index was calculated. Finally, Rubin’s rules were used to integrate the c-index values from each dataset to compute the final result. In the figure, yellow-green represents the data used for training the model, orange represents the validation data, and pink represents the data used for testing post-training

---

## [Decision Letter · Decision Letter 1]

27 Mar 2024

PONE-D-23-35647R1The Potential of the Transformer-based Survival Analysis Model, SurvTrace, for Predicting Recurrent Cardiovascular Events and Stratifying High-risk Patients with Ischemic Heart DiseasePLOS ONE

Dear Dr. Kodera,

Thank you for submitting your manuscript to PLOS ONE. After careful consideration, we feel that it has merit but does not fully meet PLOS ONE’s publication criteria as it currently stands. Therefore, we invite you to submit a revised version of the manuscript that addresses the points raised during the review process.

We look forward to receiving your revised manuscript.

Kind regards,

Marcelo Arruda Nakazone, M.D., Ph.D.

Academic Editor

PLOS ONE

Journal Requirements:

**Additional Editor Comments:**

Previous comments have been considered; nevertheless, the manuscript requires minor revisions.

Reviewers' comments:

Reviewer's Responses to Questions

**Comments to the Author**

1. If the authors have adequately addressed your comments raised in a previous round of review and you feel that this manuscript is now acceptable for publication, you may indicate that here to bypass the “Comments to the Author” section, enter your conflict of interest statement in the “Confidential to Editor” section, and submit your "Accept" recommendation.

Reviewer #1: All comments have been addressed

Reviewer #2: (No Response)

2. Is the manuscript technically sound, and do the data support the conclusions?

Reviewer #1: Yes

Reviewer #2: Yes

3. Has the statistical analysis been performed appropriately and rigorously? 

Reviewer #1: I Don't Know

Reviewer #2: Yes

4. Have the authors made all data underlying the findings in their manuscript fully available?

Reviewer #1: Yes

Reviewer #2: No

5. Is the manuscript presented in an intelligible fashion and written in standard English?

Reviewer #1: Yes

Reviewer #2: Yes

6. Review Comments to the Author

Reviewer #1: Thank you for addressing my concerns.

The outcome occurs in 8% of the population.

How would you manage this imbalance in outcomes?

Reviewer #2: The authors have addressed all my concerns. I just have two suggestions.

1. The authors should consider removing the word "retrospectively" and replacing "post-index" with "post-index event" in the following sentence (in the abstract):

"Each patient’s initial PCI at our hospital was designated as the index procedure, and a composite of major adverse cardiovascular events (MACE) was retrospectively monitored for up to two years post-index."

2. The authors should also consider replacing "selective bias" with "selection bias."

7. PLOS authors have the option to publish the peer review history of their article (what does this mean?). If published, this will include your full peer review and any attached files.

Reviewer #1: **Yes: **Ang Yee Gary

Reviewer #2: No

---

## [Author Response · Author response to Decision Letter 1]

30 Apr 2024

Reviewer #1: Thank you for addressing my concerns.

The outcome occurs in 8% of the population.

How would you manage this imbalance in outcomes?

Thank you for your valuable comments regarding the imbalanced data on outcomes. We appreciate your input, as the issue of imbalanced data is one of the critical challenges in developing machine learning models.

In this study, we did not employ oversampling or undersampling methods to address the imbalanced data for the following reasons:

Undersampling method

The undersampling method reduces the number of majority class instances to match the level of minority class instances, resulting in a decrease in the overall sample size. SurvTRACE, the analysis method used in this study, is derived from the Transformer and requires a larger sample size. We were concerned that a reduction in sample size might impact the performance of SurvTRACE; therefore, we decided against adopting the undersampling method.

Oversampling method and SMOTE

The oversampling method and SMOTE generate synthetic data by referring to minority class instances. In this study, we used a multiple imputation method to handle missing values and generate pseudo-complete data for analysis. We were concerned that creating additional synthetic data based on the already imputed data might result in a noisier dataset. Consequently, we decided against adopting these methods. Moreover, the report by Wallace et al. (2011 IEEE 11th International Conference on Data Mining, Vancouver, BC, Canada, 2011, pp. 754-763, doi: 10.1109/ICDM.2011.33.) indicates that applying SMOTE to imbalanced datasets with an outcome incidence rate of 5-10% and a dimensionality exceeding 100 may not lead to significant improvements in accuracy. The outcome incidence rate in our study is 8%, and the dimensionality is 171, which falls within the range indicated in Wallace et al.'s report.

Based on the above reasons, we decided not to adopt oversampling or undersampling methods for the imbalanced data in this study and instead performed the analysis using the original dataset as is.

However, as you have rightly pointed out, addressing the imbalanced data problem is crucial for improving the performance of machine learning models. In future research, we will strive to explore more effective methods for dealing with imbalanced data and aim to enhance the accuracy of the models.

We sincerely appreciate your valuable comments, which we believe will contribute to the improvement of our research.

Reviewer #2: The authors have addressed all my concerns. I just have two suggestions.

1. The authors should consider removing the word "retrospectively" and replacing "post-index" with "post-index event" in the following sentence (in the abstract):

"Each patient’s initial PCI at our hospital was designated as the index procedure, and a composite of major adverse cardiovascular events (MACE) was retrospectively monitored for up to two years post-index."

2. The authors should also consider replacing "selective bias" with "selection bias."

We greatly appreciate your valuable suggestions for improving our manuscript. We have carefully considered your comments and made the following changes:

Regarding your first suggestion, we agree that removing the word "retrospectively" and replacing "post-index" with "post-index event" will enhance the clarity of the sentence. We have revised the sentence in the abstract as follows:

Original (Page 1, Lines 12-14): "Each patient's initial PCI at our hospital was designated as the index procedure, and a composite of major adverse cardiovascular events (MACE) was retrospectively monitored for up to two years post-index."

Revised (Page 1, Lines 12-14): "Each patient's initial PCI at our hospital was designated as the index procedure, and a composite of major adverse cardiovascular events (MACE) was monitored for up to two years post-index event."

As per your second suggestion, we have replaced the term "selective bias" with "selection bias" throughout the manuscript to ensure the correct terminology is used.

Original (Page 19, Lines 302-304): "Despite this thorough tracking, some events might have been overlooked as a result of patients relocating or transferring to other hospitals, potentially leading to selective bias."

Revised (Page 19, Lines 302-304): "Despite this thorough tracking, some events might have been overlooked as a result of patients relocating or transferring to other hospitals, potentially leading to selection bias."

We believe that these changes, based on your insightful comments, will significantly improve the quality and readability of our manuscript. We sincerely appreciate the time and effort you have taken to review our work and provide such constructive comments.

Thank you once again for your valuable suggestions.

---

## [Decision Letter · Decision Letter 2]

13 May 2024

The Potential of the Transformer-based Survival Analysis Model, SurvTrace, for Predicting Recurrent Cardiovascular Events and Stratifying High-risk Patients with Ischemic Heart Disease

PONE-D-23-35647R2

Dear Dr. Kodera,

We’re pleased to inform you that your manuscript has been judged scientifically suitable for publication and will be formally accepted for publication once it meets all outstanding technical requirements.

Kind regards,

Marcelo Arruda Nakazone, M.D., Ph.D.

Academic Editor

PLOS ONE

Additional Editor Comments (optional):

Reviewers' comments:

Reviewer's Responses to Questions

**Comments to the Author**

1. If the authors have adequately addressed your comments raised in a previous round of review and you feel that this manuscript is now acceptable for publication, you may indicate that here to bypass the “Comments to the Author” section, enter your conflict of interest statement in the “Confidential to Editor” section, and submit your "Accept" recommendation.

Reviewer #1: All comments have been addressed

Reviewer #2: All comments have been addressed

2. Is the manuscript technically sound, and do the data support the conclusions?

Reviewer #1: Yes

Reviewer #2: Partly

3. Has the statistical analysis been performed appropriately and rigorously? 

Reviewer #1: I Don't Know

Reviewer #2: Yes

4. Have the authors made all data underlying the findings in their manuscript fully available?

Reviewer #1: Yes

Reviewer #2: No

5. Is the manuscript presented in an intelligible fashion and written in standard English?

Reviewer #1: Yes

Reviewer #2: Yes

6. Review Comments to the Author

Reviewer #1: Please kindly add unbalanced dataset as one of the limitation as this can help improves the manuscript.

Reviewer #2: The authors have addressed all my comments. As mentioned in my earlier comments, the most significant limitation of this study is that it relied on data from a single institution, which may make the authors unable to identify outcome events when patients seek care in other hospitals. The authors have made efforts to address this limitation. I am not fully confident that these measures will significantly eliminate the bias created by this situation. Barring this limitation, the article is technically sound.

7. PLOS authors have the option to publish the peer review history of their article (what does this mean?). If published, this will include your full peer review and any attached files.

Reviewer #1: **Yes: **Dr Ang Yee Gary

Reviewer #2: No

---

## [Editor Report · Acceptance letter]

16 May 2024

PONE-D-23-35647R2 

PLOS ONE

Dear Dr. Kodera, 

I'm pleased to inform you that your manuscript has been deemed suitable for publication in PLOS ONE. Congratulations! Your manuscript is now being handed over to our production team.

Kind regards, 

on behalf of

Professor Marcelo Arruda Nakazone 

Academic Editor

PLOS ONE